# Predictive Value of the Advanced Lipoprotein Profile and Glycated Proteins on Diabetic Retinopathy

**DOI:** 10.3390/nu14193932

**Published:** 2022-09-22

**Authors:** Josep Julve, Joana Rossell, Eudald Correig, Marina Idalia Rojo-Lopez, Nuria Amigó, Marta Hernández, Alicia Traveset, Marc Carbonell, Nuria Alonso, Didac Mauricio, Esmeralda Castelblanco

**Affiliations:** 1Institut d’Investigació Biomèdica Sant Pau (IIB Sant Pau), 08041 Barcelona, Spain; 2CIBER de Diabetes y Enfermedades Metabólicas Asociadas, Instituto de Salud Carlos III, 28029 Madrid, Spain; 3Department of Endocrinology & Nutrition, Hospital de la Santa Creu i Sant Pau, 08025 Barcelona, Spain; 4Department of Biostatistics, Rovira i Virgili University, 43201 Reus, Spain; 5Biosfer Teslab, SL, 43201 Reus, Spain; 6Metabolomics Platform, Institute of Health Research Pere Virgili (IISPV) and Rovira i Virgili University (URV), 43204 Reus, Spain; 7Department of Endocrinology & Nutrition, Arnau de Vilanova University Hospital and Lleida Biomedical Research Institute (IRBLleida), 25198 Lleida, Spain; 8Department of Ophthalmology, Arnau de Vilanova University Hospital and Lleida Biomedical Research Institute (IRBLleida), 25198 Lleida, Spain; 9Department of Ophthalmology, Germans Trias i Pujol University Hospital, 08916 Badalona, Spain; 10Department of Endocrinology & Nutrition, Germans Trias i Pujol University Hospital, 08916 Badalona, Spain; 11Department of Medicine, Autonomous University of Barcelona, 08916 Barcelona, Spain; 12Faculty of Medicine, University of Vic (UVIC/UCC), 08500 Vic, Spain; 13Unitat de Suport a la Recerca Barcelona, Institut Universitari d’Investigació en Atenció Primària Jordi Gol i Gurina (IDIAP Jordi Gol), 08007 Barcelona, Spain; 14Endocrinology, Metabolism and Lipid Research Division, Department of Internal Medicine, Washington University School of Medicine, St. Louis, MO 63110, USA

**Keywords:** diabetic retinopathy, lipoproteins, glycoproteins, triglycerides, remnants, nuclear magnetic resonance spectroscopy, type 1 diabetes, type 2 diabetes

## Abstract

This study aimed to assess whether the advanced characteristics of serum lipoprotein subclasses could better predict the risk of developing diabetic retinopathy (DR) and its severity compared to other established risk factors in subjects with type 1 (T1D) and type 2 (T2D) diabetes. This observational, cross-sectional substudy analyzed DR-related data from 309 T1D and 264 T2D subjects. The advanced lipoprotein and glycoprotein profile was determined by nuclear magnetic resonance (NMR) spectroscopy (Liposcale test). NMR analysis of lipoproteins revealed that T1D subjects with DR showed standard non-HDL particles, despite higher IDL lipid concentrations. Notably, IDL lipids were elevated in T1D subjects with worsened DR. VLDL and LDL were smaller, whereas HDL triglycerides were increased in DR compared with non-DR. On the other hand, the T2D subjects with DR showed altered characteristics in the LDL fraction, mainly revealed by a significant decrease in smaller LDL and a reduction in LDL-C. Moreover, the glycoprotein profile did not reveal significant changes among DR groups, regardless of the type of diabetes. However, lipoprotein characteristics and glycoproteins unveiled by NMR analysis did not improve the predictive value of conventional lipids or other traditional, well-established biomarkers of DR in our cohorts.

## 1. Introduction

Diabetic retinopathy (DR) is one of the most frequent microvascular complications of diabetes, affecting over one-third of diabetic subjects [1], and the most common cause of preventable blindness in working-age adults [2,3]. Regular ophthalmological examinations allow for the early detection of DR in diabetic subjects, leading to the initiation of treatments and significantly reducing the risk of developing clinical diabetic macular edema or early proliferative DR and thus blindness.

Compelling evidence indicates a link between DR and cardiovascular (CV) risk [4]. First, the severity and the progression of DR have been associated with an increased risk of CV disease [5,6,7,8,9]. Second, a series of different clinical studies have suggested a significant relationship between DR and peripheral arterial disease [10], vascular disease [11], coronary artery calcium accumulation [12], and arterial stiffness [13]. Third, the burden of atherosclerotic carotid plaques has been related to advanced signs of DR in subjects with type 1 diabetes (T1D) [14,15] and type 2 diabetes (T2D) [16,17,18,19].

Although the mechanisms underlying such relationships are still poorly understood, data suggest that diabetic dyslipidemia could be a contributing factor in both macrovascular and microvascular complications [20]. Indeed, the notion that diabetic dyslipidemia contributes to atherosclerosis is well established [21,22,23]. Likewise, an association between the development of DR and diabetic dyslipidemia has also been revealed in different clinical studies in both T1D [24,25] and T2D [26,27,28,29].

Despite this body of evidence, a recent study showed no significant changes in serum lipids in T1D subjects with DR [14]. This may, at least in part, be explained by the fact that the subjects with more severe DR received more medications for dyslipidemia than those with milder DR [14]. In addition, conventional lipid analysis does not capture critical lipoprotein abnormalities that may predispose to either microvascular or macrovascular damage.

Of note, we recently found hidden derangements in the lipoprotein profile of T1D using a nuclear magnetic resonance (NMR) spectroscopy approach in apparently normolipidemic T1D subjects [30]. This approach also allowed the identification of lipoprotein characteristics associated with preclinical atherosclerosis in T1D subjects [31] and newly diagnosed T2D subjects [23]. However, the added predictive potential shown by the advanced lipoprotein profile to diagnose DR has not yet been reported. Thus, we tested the hypothesis that quantitative characteristics of lipoproteins (and glycoproteins) determined by NMR could improve the predictive potential of currently used biomarkers for diagnosing DR in two well-characterized cohorts of T1D and T2D subjects.

## 2. Materials and Methods

This was a cross-sectional substudy in subjects with T1D and T2D from cohorts previously recruited from the diabetic outpatient clinics at two university hospitals in a northeastern region of Spain (Catalonia) to study the relationship between carotid atherosclerosis and DR [14,16]. In this substudy, we analyzed a subpopulation of subjects who had a complete ophthalmological evaluation together with a sufficient serum sample for lipoprotein and glycoprotein determinations. For both T1D and T2D groups, inclusion criteria were as follows: diabetes for at least one year; normal renal function (estimated glomerular filtration rate (eGFR) >60 mL/min/1.73 m^2^); no previous CV disease, defined as any form of clinical coronary heart disease, stroke, or peripheral vascular disease; and any form of diabetic foot disease. Furthermore, subjects were aged >18 years for T1D and 40 to 75 years for T2D. We excluded patients with a urine albumin-creatinine ratio (ACR) ≥30 mg/g and any conditions that precluded a complete eye examination.

### 2.1. Clinical Assessment

For each subject, age, sex, weight, height, body mass index and waist circumference were obtained by standard methods. Serum and spot urine samples were collected in the fasting state, and all tests were performed using standard laboratory methods as previously described [16]. The eGFR was calculated according to Chronic Kidney Disease Epidemiology Collaboration equation [32]. The fatty liver index (FLI) was calculated as: (e^0.953 × loge(triglycerides) + 0.139 × BMI + 0.718 × loge(GGT) + 0.053 × (waist circumference) − 15.745^)/(1 + e^0.953 × loge(triglycerides) + 0.139 × BMI + 0.718 × loge(GGT) + 0.053 × (waist circumference) − 15.745^) × 100 [33]. Hypertension and dyslipidemia were defined as a recorded clinical diagnosis and current use of antihypertensive and lipid-lowering medication.

The study was conducted in accordance with the tenets of the Declaration of Helsinki, and approval was obtained from the ethics committees of both participating centers: the Arnau de Vilanova University Hospital (Lleida) (CEIC-7/2011, August 2011) and the Germans Trias i Pujol University Hospital (Badalona) (PI-15-147, December 2015). Written informed consent was provided by all participants.

### 2.2. Assessment of Diabetes and Presence and Severity of Diabetic Retinopathy

Diabetes was defined and classified according to the American Diabetes Association criteria [34]. To assess the presence and severity of DR, the patients underwent a complete eye evaluation by experienced ophthalmologists (AT and MC) according to an international clinical DR consensus [35]. Participants underwent pupil dilation, followed by multifield stereoscopic retinal photography for both eyes: Early Treatment for Diabetic Retinopathy Study (ETDRS) field 1, which is centered on the optic disc, and field 2, which is centered on the fovea. DR was classified into five stages according to the ETDRS classification [23]: (1) no apparent retinopathy, (2) mild nonproliferative retinopathy (NPDR), (3) moderate NPDR, (4) severe NPDR, and (5) proliferative diabetic retinopathy (PRD). Diabetic subjects with DR were reclassified into 2 stages (mild [ETDRS stages 1–2] or advanced [ETDRS stages 3–4] DR). For the analysis, we used the ophthalmological variables obtained for the right eye if both eyes had the same DR grade; otherwise, we used the variables from the eye with the highest DR grade [36].

### 2.3. Nuclear Magnetic Resonance (NMR) Molecular Profiling

The NMR analysis, which includes the lipoprotein profile based on the Liposcale test [37] and the glycoprotein profile [38], was carried out at Biosfer Teslab (Reus, Spain). Serum samples were thawed overnight and prepared for NMR analyses. To detect larger molecules, such as lipoproteins and glycoproteins, high-resolution ^1^H-NMR spectroscopy was conducted on a Bruker 600 MHz spectrometer by LED diffusion (Diff) experiments, running at 37 °C in quantitative conditions. The three main lipoprotein types (very low-density lipoprotein [VLDL], low-density lipoprotein [LDL], and high-density lipoprotein [HDL] were assessed using the Liposcale test, which was used to obtain the composition, mean size, and number of lipoprotein particles of nine subtypes, i.e., large, medium, and small VLDL, LDL and HDL. The composition of IDL-P, intermediate-density lipoprotein particles, was also assessed [37]. The variable percentage associated with particle distribution was calculated as the number of small particles divided by the total and multiplied by 100 for VLDL, LDL, and HDL. Composition ratios were obtained by dividing the levels of triglyceride (TG) by cholesterol (C) in each fraction. From the same NMR spectra, we obtained the concentration of the glycoproteins, i.e., the acetyl groups of *N*-acetylglucosamine and *N*-acetyl galactosamine (GlycA) and acetyl groups of *N*-acetylneuraminic acid (GlycB) and their height/width ratios (H/W GlycA and H/W GlycB), as previously described [39].

### 2.4. Statistical Analysis

Continuous variables were tested for normality using the Kolmogorov–Smirnov test. Data are presented as means and standard deviation (SD) for continuous variables, and *n* and percentages for categorical variables. Differences between groups were analyzed using the nonparametric Mann–Whitney test or Student’s parametric t-test for continuous variables and the chi-square test or Fisher’s exact test for categorical variables.

We used random forests and regularized logistic regression to construct the prediction models for the presence and severity of DR. For the prediction models of DR severity, we used random forests and multinomial regression. Model parameters were adjusted using 5-fold cross-validation, and the models were trained on 80% of the dataset and tested on the remaining 20% of data. We trained several models of each type (regression or random forest) and chose the best performing in each case. To assess the effect of the predictive variables on the target, we carried out a dual approach. Random forests and boosting have proved to be more accurate models for such complex scenarios [40], and, as an added advantage, they allowed us to assess the relative importance of each variable via out-of-bag accuracy. However, the downside is that random forests did not allow us to evaluate whether each variable has a protective or harmful effect. For this, we used the result of the regularized logistic regression, through which we could evaluate the direction of the effect via the sign of the coefficient. Since this is the only parameter we were interested in, confidence intervals and *p*-values were not reported for the multivariate models.

The traditional models were performed with the following variables: sex, age, waist circumference, hypertension, DM duration, HbA1c, eGFR, ACR, high-sensitivity C-reactive protein (hs-CRP), total cholesterol (Total C), HDL-C, LDL-C, and TG. The Liposcale model included the same variables as the traditional model, exchanging Total C, HDL-C, LDL-C and TG for the values determined by NMR, and adding variables from NMR that were significantly different between groups. To assess model performance, we computed the accuracy and the area under the ROC curve (AUROC) for all models. Statistical analyses were performed using the R statistical software version 4.0.5 [41].

## 3. Results

### 3.1. Clinical Characteristics in Diabetic Subjects with Diabetic Retinopathy

Overall, 573 participants were included in this substudy: 309 with T1D and 264 with T2D. After ophthalmological examination, out of the 309 subjects with T1D, 48 were classified as having advanced DR, 80 with mild DR, and 181 were without any DR. The 264 subjects with T2D included 75 with advanced DR, 50 with mild DR, and 139 without DR (Appendix A).

Subjects with DR had a longer diabetes duration regardless of diabetes type and poorer glycemic control (higher HbA1c values) (Table 1). They also had an increased incidence of subclinical carotid atherosclerosis and higher systolic blood pressure (sBP), despite receiving more medication to treat hypertension (Table 1). Of note, all the abovementioned clinical characteristics showed a trend to be worse in subjects with advanced DR (Appendix A).

Analysis by diabetes type according to DR status showed that T1D subjects with DR were older (+4.9 years) than those without DR, with a higher prevalence of dyslipidemia and higher serum TG values, despite receiving lipid-lowering treatment more frequently (Table 1). Moreover, T1D subjects with DR had higher BMI than those without DR, with concomitantly higher FLI values (Table 1). Of note, the FLI value increased according to the severity of DR (Appendix A). These patients also had lower eGFR values (Table 1). As expected, microalbuminuria was significantly elevated in the group of T1D subjects with advanced DR (Appendix A). Regarding oral antiplatelet treatment, T1D subjects with advanced DR were more frequently treated with these drugs (Appendix A).

On the other hand, the T2D subjects with DR had higher HbA1c values than those without DR (Appendix A). No other clinical parameters were specifically different between T2D subjects with and without DR, except for microalbuminuria, which was increased only in DR patients (Table 1), particularly in subjects with advanced DR (threefold) compared to non-DR subjects (Appendix A). Likewise, urine ACR was significantly higher (threefold) in this group (Table 1), and more apparent in those T2D subjects with advanced stages of DR (Appendix A). T2D subjects with DR also more frequently received antiplatelet agents, particularly those with advanced DR (Table 1 and Appendix A). The clinical characteristics of T1D and T2D subjects distributed according to sex are shown in Appendix A.

### 3.2. Advanced Lipoprotein and Glycoprotein Characteristics

The advanced lipoprotein NMR analyses revealed a different pattern of changes between subjects with T1D and subjects with T2D when comparing DR and non-DR patients (Table 2). In the T1D group, the values of non-HDL particles in DR patients did not differ from non-DR patients, despite having higher IDL lipid concentrations. The IDL lipids were increased according to DR severity (Appendix A). The advanced analysis also uncovered changes in other differential lipoprotein characteristics, including the size of VLDL and LDL particles, which were smaller than average on subjects with DR (Table 2). In line with this, the concentration of small LDL was increased in DR compared with non-DR subjects. Similarly, the concentration of smaller VLDL was also increased, though marginally, in the DR group of T1D subjects. Finally, the concentration of HDL-TG was significantly increased in DR compared with non-DR.

In contrast with the abovementioned lipoprotein characteristics, subjects with T2D and DR had a lower concentration of LDL-C than non-DR T2D subjects that was accompanied by a significant decrease in the concentration of small LDL (Table 2). Another different characteristic was the significant elevation in the medium HDL in the DR patients that was not seen in subjects with T1D.

Serum glycoproteins in DR patients did not differ from those in non-DR subjects in any of the diabetes subgroups (Table 2).

### 3.3. Contribution of Advanced Lipoprotein Characteristics and Glycoproteins to DR Prediction

We used two models to evaluate whether adding advanced lipoprotein characteristics and glycated proteins (i.e., Liposcale model) to the usual clinical risk factors improved the capacity to predict the presence of DR compared to routine lipid parameters only (i.e., Traditional model).

In T1D, the most important variable was diabetes duration in both the Traditional and the Liposcale models (Figure 1a,b). Regarding the Traditional model, other clinically relevant variables (i.e., hypertension, HbA1c, and waist) were revealed as predictors of DR (Figure 1c). The Liposcale model showed that high IDL-TG and low H/W GlycB were significantly associated with the presence of DR (Figure 1d). However, the Liposcale model did not improve upon the Traditional one, with accuracy values of 0.74 and an AUROC of 0.79, whereas the accuracy and AUROC values provided by the Liposcale model were 0.73 and 0.80. To identify a possible masking effect from diabetes duration, we categorized the variable in tertiles and repeated the analysis with both models; however, even this strategy did not improve the predictive potential of advanced variables (Appendix A).

In T2D, ACR, diabetes duration and HbA1c were the most important variables to predict DR (Figure 1e,f). In addition to these, the regression analyses also found other clinically relevant variables, independent of the model used (i.e., Traditional and Liposcale) (Figure 1g,h). No additional lipoprotein variables were predictive of DR on the Liposcale model. As with T1D, the accuracy and AUROC values calculated using the Liposcale model (accuracy: 0.75; AUROC: 0.82) did not differ from the Traditional model (accuracy: 0.74; AUROC: 0.83).

### 3.4. Contribution of Advanced Lipoprotein Characteristics and Glycoproteins to DR Severity Prediction

In order to determine the variables that predict the severity of DR, we evaluated mild DR vs. severe DR and no DR vs. severe DR, analyzed by random forest and regularized logistic regression, and DR stages (no, mild, and severe DR) analyzed by random forest and multinomial analysis. Further, every approximation was evaluated using the Traditional and Liposcale models.

#### 3.4.1. Mild DR vs. Severe DR

In T1D, there was agreement between both the Traditional and Liposcale models on the most important variables for predicting severe DR (Appendix A), i.e., longer diabetes duration, hypertension, and high HbA1c (Appendix A). In the Liposcale model, none of the Liposcale variables or glycoproteins were significant. The accuracy and AUROC did not differ between the Traditional model (0.84 and 0.80, respectively) and the Liposcale model (0.83 and 0.80, respectively). On the other hand, in the T2D group, the most important variables to predict DR severity were ACR, diabetes duration, and HbA1c (Appendix A). In addition to the high ACR, longer duration of DM, and high HbA1c, the regression models also uncovered other clinical variables (i.e., waist, HDL-C) potentially contributing to DR prediction (Appendix A). Of note, some of the Liposcale variables in the Liposcale model, i.e., the addition of LDL-TG and to a lesser extent HDL-TG, GlycB and IDL-TG, were shown to predict DR. Nevertheless, the calculated accuracy and AUROC values (accuracy: 0.72; AUROC: 0.81) were lower than in the Traditional model (accuracy: 0.78; AUROC: 0.80).

#### 3.4.2. No DR vs. Severe DR

In the T1D group, in both the Traditional and Liposcale models, the most important variable to predict severe DR was diabetes duration (Appendix A). In addition, the regression models unveiled other clinical variables including hypertension, HbA1c, and waist (Appendix A). Of note, the IDL-TG was found to be a predictive variable in the Liposcale model; however, the accuracy and AUROC values did not differ between models (Liposcale model: accuracy 0.81 and AUROC 0.85; Traditional model: accuracy 0.84 and AUROC 0.85). Regarding the T2D group, ACR, diabetes duration and HbA1c were the most important variables identified in both models (Appendix A). The regression models also identified other clinical variables (i.e., HDL-C, waist) with DR predictive potential (Appendix A). In this context, the Liposcale model identified some of the Liposcale variables (i.e., HDL-C and, to a lesser extent HDL-TG), which showed a mild improvement in the calculated accuracy and AUROC values in the Liposcale model (accuracy: 0.80; AUROC: 0.85) compared with the Traditional model (accuracy: 0.75; AUROC: 0.86).

#### 3.4.3. DR Stages

According to the random forest analysis, in the T1D group, the most important variable to predict DR stage (no, mild, or severe DR) in both the Traditional and Liposcale models was again diabetes duration (Appendix A). The assessment of the multinomial models assigned an accuracy value of 0.67 to the Traditional model and 0.63 to the Liposcale model. Conversely, in the T2D group, the most critical variables predicting DR stage were the ACR, diabetes duration, and HbA1c in both the Traditional and Liposcale models (Appendix A). There was no improvement in the prediction of severity after adding Liposcale variables when using the multinomial models; accuracy values were 0.67 for both the Traditional and Liposcale models.

## 4. Discussion

The relationship between either advanced lipoprotein characteristics or glycoproteins with atherogenicity has recently been reported in both subjects with T1D [31,42] and T2D [23,39]; however, their contribution to DR has not been previously evaluated. Therefore, we focused on assessing the relationship between the characteristics of circulating lipoprotein and glycoprotein and DR in T1D and T2D subjects, and whether they contributed to a better prediction of the presence and severity of DR. Our data showed that neither lipoprotein parameters nor glycoproteins determined by NMR enhanced the prediction of DR presence or severity in our T1D and T2D cohorts. However, consistently with previous reports, diabetes duration and poorer glycemic control were among the main risk factors for DR presence and severity [43]. The predominant effect shown in the models by traditional variables, such as diabetes duration, HbAc1, or elevated albuminuria-to-creatinine ratio, could hint that they are obscuring the predictive potential of advanced lipoprotein and glycoprotein values; however, the categorization in tertiles of one of them, i.e., diabetes duration, did not improve the predictive potential of such advanced variables.

Several studies have reported a link between DR and impaired renal function [44,45,46,47,48], with an elevated incidence of DR in subjects with diabetic nephropathy [49]. In the present study, only subjects with normal renal function were included. This is an important point, as DR prediction could be performed without confounding established kidney microvascular disease. In this regard, both urine ACR and microalbuminuria concomitantly fell into the normal range in the different groups of diabetic patients, regardless of DR. Interestingly, our analysis revealed an association between subclinical values of kidney disease and DR in T2D subjects. Thus, even in the absence of renal dysfunction, urine ACR was identified as a strong predictor of DR, for both T1D and T2D subjects. Supporting this, microalbuminuria was concomitantly elevated in T1D and T2D subjects with signs of severe DR.

DR has been related to subclinical atherosclerosis plaque in both T1D [14] and T2D [9,19,50]. Apart from its role as a risk factor for the development of atherosclerosis and CV disease, accumulating evidence suggests that diabetic dyslipidemia may play a role in DR progression [51,52]. In this regard, both TG elevations and HDL-C reductions have been found to be associated with DR [53]. However, our data revealed HDL-C elevations in T2D subjects with either DR presence or severity. Noteworthy, this finding was consistent with previous data [54], showing that higher HDL-C levels were directly associated with the presence and severity of DR in T2D subjects. To confirm this notion in our study, we performed a subanalysis aimed at assessing the proportion of both T1D and T2D subjects with DR or severity in three HDL-C concentration ranges (<30, 30–60, and >60 mg/dL), as defined in previous work [54]. Our data further confirmed that the percentage of T2D subjects with DR or advanced signs of DR was elevated in the group of T2D subjects with higher HDL-C concentrations (>60 mg/dL). Conversely, the HDL-C levels in T1D subjects did not differ according to DR presence or severity. Such controversy could be at least in part explained by the very distinct metabolic context of T1D compared with T2D. Furthermore, increased concentrations of small-dense LDL, which is considered a risk factor for future CV events, may predict advanced stages of DR [55]. However, in this study, the models that included the advanced lipoprotein fractions did not improve the prediction of DR presence or severity compared to the models with the traditional risk factors. Despite this, the serum concentration of smaller LDL particles in our T1D subjects with DR was slightly increased compared with those without DR; however, this lipoprotein change was only marginal when the DR group of T1D subjects were further differentiated into two groups according to the degree of DR severity, and thus it is considered of minor clinical significance. On the other hand, the advanced lipoprotein analysis uncovered a significant decrease in the concentration of total LDL-C in T2D patients with DR, which was at least in part due to a reduction in the smaller fraction of LDL. Although we do not have an explanation for this intriguing decrease in the smaller LDL, it was not attributed to changes in the relative proportion of subjects taking hypocholesterolemic medication (i.e., statins).

DR may also be predicted by serum elevations of cholesterol remnants, i.e., VLDL remnants, independently of other risk factors [56]. Supporting this concept, apolipoprotein B-containing lipoproteins might be selectively retained in the capillary beds of the retina as it is thought to occur during atherosclerotic coronary artery disease [57]. In our study, the VLDL remnants, which include IDL, were significantly elevated in T1D subjects with DR, with such elevations increasingly higher with DR severity. Despite this, our algorithms revealed that IDL lipids did not improve DR prediction in T1D subjects.

Elevated cholesterol remnants have recently been reported as a potential risk factor for DR in T2D subjects [58]; however, we did not observe elevations in the serum concentrations of this lipoprotein class in our T2D subjects with DR. The reason for the inconsistent results between the former study and our data is currently unknown, but it could be partly explained by the different approaches used to estimate the remnant cholesterol concentrations in each study. In this respect, our NMR analysis provided a more accurate estimation of serum cholesterol remnant concentrations than that used in the previous studies [56,58], which simply consisted of subtracting the LDL-C and HDL-C moieties from Total-C. Like in the case of T1D, IDL lipids did not contribute to improving predicting DR in our cohort of subjects with T2D.

Triglyceride-rich HDL, which is considered a risk factor for CV disease [27] was found increased only in the DR group of T1D subjects. This parameter is also a hallmark of T1D at onset [59], and is favorably influenced by an intensive optimization of glycemic control and improved CV risk in T1D subjects [30]. Insofar as DR progression parallels atherosclerosis development in T1D [14], it could be hypothesized that elevated triglyceride-rich HDL might be considered a potential candidate risk factor of DR. However, the relative weight of HDL-TG to improve DR prediction in this cohort was marginal.

Systemic chronic inflammation, which is frequently found in subjects with diabetes [60], profoundly influences the development and progression of DR [61]. Of note, circulating NMR-derived glycoproteins have been recently linked to inflammation [38,39]. Considering that the glycoprotein profile has been recently associated with atherogenic dyslipidemia in subjects with T2D [39], we assessed their added predictive value along with advanced characteristics of lipoproteins over other traditional risk factors of DR. Our data showed that glycoprotein values in the DR groups did not significantly differ from those without DR, regardless of the type of diabetes mellitus. The glycoprotein characteristics did not improve the predictive models for the presence or the severity of DR in any of the two diabetic scenarios analyzed. The relatively high prevalence of atherosclerosis plaque, which is considered a sign of chronic inflammation, in subjects with advanced DR suggested an increased inflammatory status in this group. Consistently with this, hs-CRP was directly related to DR in our cohorts.

One of the main strengths of this substudy was the exclusion of both T1D and T2D subjects with advanced nephropathy, a condition that often coexists with DR and that may affect the metabolism of some of the assessed circulating variables [55]. As such, a major confounding factor was eliminated, providing a better evaluation of lipoprotein characteristics uncovered by NMR analysis and DR. However, our study also had some flaws. First, the observational, cross-sectional nature of this substudy did not allow us to ensure complete control of all the potential (still unknown) confounding factors. Second, the causal nature of the relationship between atherosclerotic plaque formation and increased risk for DR could not be proven. Last, the current study did not assess the effect of lipid-lowering or antihypertensive medications on DR risk.

## 5. Conclusions

Although our NMR spectroscopy unveiled differential hidden characteristics of lipoproteins in T1D and T2D subjects with DR, it failed to improve the predictive value provided by other traditional factors. Neither the presence nor severity of DR was linked to changes in the serum concentrations of glycoproteins, which have been recently established as inflammation biomarkers for CV disease. At least in part, the predictive power of diabetes duration for the presence and severity of DR over other variables in each model was so high that it could have been masking the added predictive value of advanced variables.

## Figures and Tables

**Figure 1 nutrients-14-03932-f001:**
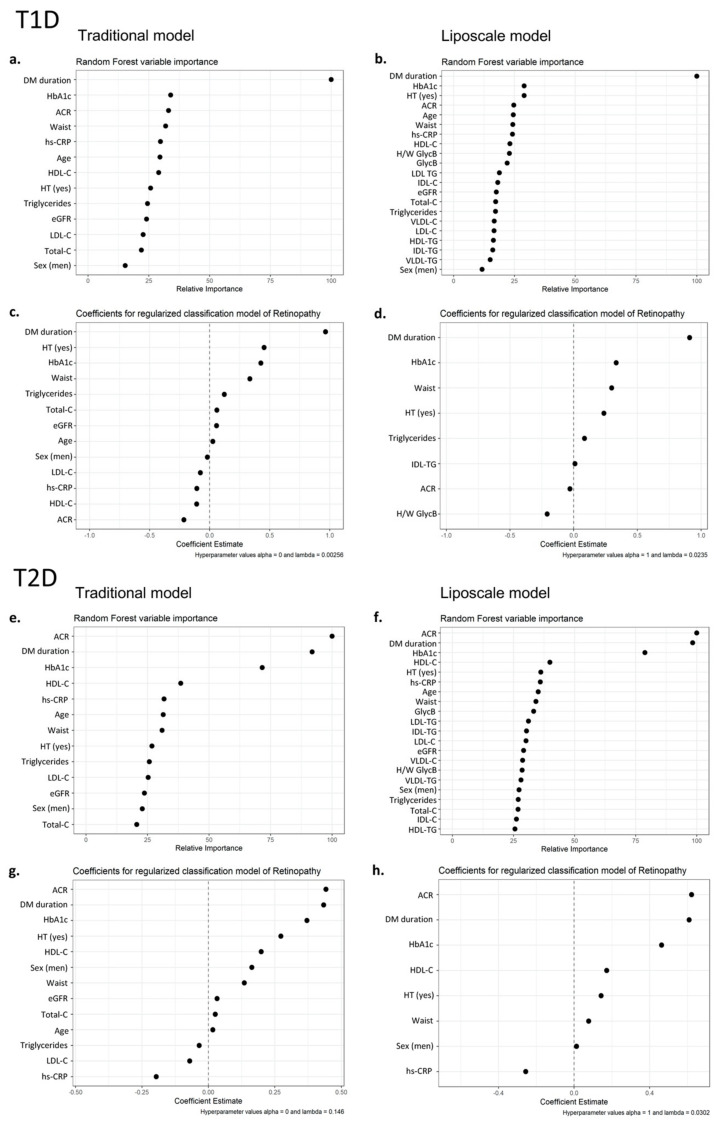
Traditional and Liposcale models to predict the presence of DR in T1D and T2D groups. In the T1D group, random forest analysis for Traditional (**a**) and Liposcale (**b**) models, and regularized logistic regression analysis for Traditional (**c**) and Liposcale (**d**) models. In the T2D group, random forest analysis for Traditional (**e**) and Liposcale (**f**) models, and regression analysis for Traditional (**g**) and Liposcale (**h**) models. ACR, albumin-to-creatinine ratio; BMI, body mass index; DM, diabetes mellitus; dBP, diastolic blood pressure; eGFR, estimated glomerular filtration rate calculated according to Chronic Kidney Disease Epidemiology Collaboration (CKD-EPI) equation; GlycB, area proportional to the concentration of the acetyl groups of acetyl groups of *N*-acetylneuraminic acid; HbA1c, glycated hemoglobin; HDL-C, high-density lipoprotein cholesterol; HDL-TG, high-density lipoprotein triglyceride; hs-CRP, high sensitive C reactive protein; H/W GlycB, height-to-width ratio of GlycB; IDL-C, intermediate-density lipoprotein cholesterol; IDL-TG, intermediate-density lipoprotein triglyceride; LDL-C, low-density lipoprotein cholesterol; LDL-TG, low-density lipoprotein triglyceride; VLDL-C, very low-density lipoprotein cholesterol; VLDL-TG, very low-density lipoprotein triglyceride.; sBP, systolic blood pressure Total-C, total cholesterol.

**Table 1 nutrients-14-03932-t001:** Clinical characteristics according to diabetes type and presence or absence of diabetic retinopathy.

Characteristics	T1D			T2D		
	Non-DR	DR	*p*-Value	Non-DR	DR	*p*-Value
	*n* = 181	*n* = 128		*n* = 139	*n* = 125	
Age (years)	44.0 (10.9)	48.9 (12.1)	<0.001	57.4 (10.0)	59.5 (8.58)	0.073
Sex (women)	95 (52.5%)	68 (53.1%)	1	66 (47.5%)	60 (48.0%)	1
BMI (kg/m^2^)	25.4 (3.86)	26.8 (4.26)	0.003	31.3 (5.25)	31.8 (5.66)	0.505
Waist circumference (cm)	87.8 (11.7)	91.9 (12.9)	0.005	104 (12.1)	107 (10.6)	0.085
sBP (mmHg)	125 (17.5)	132 (18.7)	0.001	134 (15.7)	144 (20.7)	<0.001
dBP (mmHg)	74.6 (10.1)	73.8 (10.1)	0.485	76.4 (10.2)	77.0 (10.8)	0.601
Hypertension (yes)	35 (19.3%)	53 (41.4%)	<0.001	68 (48.9%)	81 (64.8%)	0.013
Dyslipidemia (yes)	68 (37.6%)	66 (51.6%)	0.020	61 (43.9%)	63 (50.4%)	0.350
Smoking:			0.596			0.496
Active smoker	47 (26.0%)	31 (24.2%)		29 (20.9%)	26 (20.8%)	
Former smoker	43 (23.8%)	37 (28.9%)		50 (36.0%)	37 (29.6%)	
DM duration (years)	17.9 (9.72)	27.2 (9.96)	<0.001	6.88 (5.48)	13.6 (9.51)	<0.001
Glucose (mg/dL)	163 (71.6)	170 (78.5)	0.398	148 (49.2)	166 (58.1)	0.010
Creatinine (mg/dL)	0.77 (0.16)	0.77 (0.14)	0.900	0.81 (0.17)	0.81 (0.17)	0.901
eGFR (mL/min/1.73 m^2^)	103 (13.9)	99.9 (14.0)	0.026	92.0 (14.5)	90.4 (14.5)	0.381
Triglycerides (mg/dL)	72.6 (29.5)	83.3 (48.4)	0.028	136 (68.9)	133 (70.7)	0.762
Total C (md/dL)	179 (28.1)	181 (34.4)	0.708	186 (36.4)	185 (36.6)	0.913
HDL-C (mg/dL)	64.3 (14.4)	62.7 (17.2)	0.378	48.1 (10.6)	52.2 (15.5)	0.013
LDL-C (mg/dL)	101 (23.1)	102 (28.1)	0.684	112 (30.6)	107 (30.2)	0.200
HbA1c (%)	7.48 (0.94)	7.88 (1.09)	0.001	7.29 (1.16)	8.36 (1.46)	<0.001
HbA1c (mmol/mol)	58.3 (10.3)	62.6 (11.9)	0.001	56.1 (12.7)	67.9 (16.0)	<0.001
hs-CRP (mg/L)	3.11 (6.88)	2.66 (3.85)	0.465	4.44 (4.47)	3.86 (5.22)	0.339
Plaque:			0.001			<0.001
Multiple plaques	21 (11.6%)	33 (25.8%)		28 (20.1%)	54 (43.2%)	
No plaque	134 (74.0%)	69 (53.9%)		69 (49.6%)	42 (33.6%)	
One plaque	26 (14.4%)	26 (20.3%)		42 (30.2%)	29 (23.2%)	
FLI	24.3 (22.1)	32.9 (26.1)	0.003	66.8 (22.7)	69.4 (23.0)	0.349
Microalbuminuria (mg/L)	9.26 (24.4)	13.9 (25.2)	0.108	11.1 (14.7)	30.1 (37.2)	<0.001
ACR (mg/g)	4.48 (11.0)	6.27 (19.4)	0.349	9.93 (18.6)	29.7 (39.6)	<0.001

Data are shown as *n* (%) for categorical variables and mean (SD) for continuous variables. ACR, albumin-to-creatinine ratio; BMI, body mass index; C, cholesterol; dBP, diastolic blood pressure; DM, diabetes mellitus; eGFR, estimated glomerular filtration rate calculated according to Chronic Kidney Disease Epidemiology Collaboration (CKD-EPI) equation; FLI, Fatty Liver Index; HbA1c, glycated hemoglobin; HDL-C, high-density lipoprotein cholesterol; hs-CRP, high-sensitivity C reactive protein; LDL-C, low-density lipoprotein cholesterol; sBP, systolic blood pressure.

**Table 2 nutrients-14-03932-t002:** Advanced lipoprotein profile according to diabetes type and presence or absence of diabetic retinopathy.

Advanced Lipoprotein Profile	T1D			T2D		
	Non-DR	DR	*p*-Value	Non-DR	DR	*p*-Value
	*n* = 181	*n* = 128		*n* = 139	*n* = 125	
VLDL-P number (nmol/L)						
Total	30.2 (14.6)	34.8 (24.1)	0.059	69.1 (45.2)	69.6 (56.4)	0.935
Large	0.81 (0.32)	0.89 (0.47)	0.081	1.63 (0.89)	1.62 (1.21)	0.949
Medium	3.11 (1.79)	3.51 (3.29)	0.213	6.26 (6.68)	6.49 (8.04)	0.804
Small	26.3 (12.7)	30.4 (20.5)	0.050	61.2 (38.6)	61.5 (48.0)	0.956
VLDL-P composition						
VLDL-C (mg/dL)	7.62 (5.65)	9.13 (8.68)	0.087	17.1 (13.4)	17.4 (15.0)	0.875
VLDL-TG (mg/dL)	43.0 (20.1)	49.1 (34.2)	0.071	98.8 (68.0)	99.6 (86.9)	0.928
VLDL-P size (nm)	42.2 (0.23)	42.1 (0.23)	0.044	42.0 (0.21)	42.0 (0.22)	0.315
LDL-P number (nmol/L)						
Total	1265 (193)	1285 (230)	0.415	1355 (252)	1276 (262)	0.013
Large	182 (29.6)	182 (31.9)	0.871	174 (31.5)	173 (35.4)	0.743
Medium	413 (106)	409 (123)	0.768	392 (129)	379 (137)	0.428
Small	669 (95.1)	694 (115)	0.046	789 (124)	724 (134)	<0.001
LDL-P composition						
LDL-C (mg/dL)	125 (19.7)	125 (23.4)	0.895	127 (25.2)	120 (26.2)	0.026
LDL-TG (mg/dL)	15.8 (4.31)	16.5 (4.76)	0.167	17.4 (4.74)	17.9 (5.45)	0.476
LDL-P size (nm)	21.1 (0.24)	21.0 (0.25)	0.043	20.8 (0.23)	20.9 (0.32)	0.010
HDL-P number (nmol/L)						
Total	32.7 (5.95)	33.0 (7.02)	0.771	27.1 (5.01)	27.7 (6.02)	0.348
Large	0.28 (0.05)	0.29 (0.05)	0.117	0.26 (0.04)	0.27 (0.05)	0.354
Medium	10.8 (2.42)	11.0 (2.67)	0.614	8.04 (1.34)	8.49 (2.15)	0.045
Small	21.6 (4.21)	21.7 (5.03)	0.908	18.8 (4.24)	18.9 (4.71)	0.734
HDL-P composition						
HDL-C (mg/dL)	65.6 (13.4)	65.5 (16.6)	0.938	49.4 (9.47)	51.1 (12.7)	0.214
HDL-TG (mg/dL)	13.8 (3.98)	14.8 (3.83)	0.032	14.0 (4.58)	14.4 (4.79)	0.471
HDL-P size (nm)	8.23 (0.06)	8.24 (0.06)	0.447	8.20 (0.07)	8.21 (0.07)	0.302
IDL-P composition						
IDL-C (mg/dL)	9.38 (4.43)	10.8 (5.03)	0.011	13.1 (4.83)	13.5 (5.37)	0.572
IDL-TG (mg/dL)	10.7 (3.37)	11.8 (3.87)	0.011	14.5 (4.13)	15.0 (4.34)	0.377
Other atherogenic variables						
Non-HDL-P (nmol/L)	1262 (198)	1287 (233)	0.332	1397 (254)	1318 (262)	0.013
Total-P/HDL-P	41.0 (10.9)	41.7 (11.3)	0.597	54.5 (15.1)	50.9 (14.9)	0.046
LDL-P/HDL-P	40.0 (10.5)	40.6 (10.8)	0.680	51.8 (14.3)	48.1 (14.1)	0.035
Total C (mg/dL)	208 (26.2)	211 (32.1)	0.366	207 (32.0)	202 (33.5)	0.245
Total TG (mg/dL)	83.3 (27.0)	92.2 (41.5)	0.034	142 (62.2)	142 (65.2)	0.983
GlycA	5.12 (1.03)	5.18 (1.13)	0.620	6.71 (1.51)	6.71 (1.51)	0.969
GlycB	2.02 (0.40)	1.99 (0.39)	0.492	2.25 (0.32)	2.30 (0.37)	0.297
H/W GlycA	17.1 (3.45)	16.9 (2.91)	0.607	21.8 (3.78)	21.9 (4.44)	0.940
H/W GlycB	4.88 (0.93)	4.76 (0.78)	0.222	5.93 (0.92)	5.94 (0.93)	0.892

Data are shown as mean (SD) for continuous variables. GlycA, area proportional to the concentration of the acetyl groups of *N*-acetylglucosamine and *N*-acetyl galactosamine; GlycB, area proportional to the concentration of the acetyl groups of acetyl groups of *N*-acetylneuraminic acid; HDL, high-density lipoprotein; HDL-C, high-density lipoprotein cholesterol; HDL-P, high-density lipoprotein particle; HDL-TG, high-density lipoprotein triglyceride; H/W GlycA, height-to-width ratio of GlycA; H/W GlycB, height-to-width ratio of GlycB; IDL, intermediate-density lipoprotein, IDL-C, intermediate-density lipoprotein cholesterol; IDL-P, intermediate-density lipoprotein particle; IDL-TG, intermediate-density lipoprotein triglyceride; LDL, low-density lipoprotein; LDL-C, low-density lipoprotein cholesterol; LDL-P, low-density lipoprotein particle; LDL-TG, low-density lipoprotein triglyceride; VLDL, very low-density lipoprotein; VLDL-C, very low-density lipoprotein cholesterol; VLDL-P, very low-density lipoprotein particle; VLDL-TG, very low-density lipoprotein triglyceride.

## Data Availability

Datasets presented in this study are available on request from the corresponding author.

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
