# Peer review of "Predictive Value of the Advanced Lipoprotein Profile and Glycated Proteins on Diabetic Retinopathy"

_nutrients, 2022, doi:10.3390/nu14193932_

Round 1
Reviewer 1 Report
This study is to demonstrate the Prediction whether the advanced characteristics of serum lipoprotein subclasses could better predict the risk of developing diabetic retinopathy (DR) and its severity compared to other established risk factors in subjects with type 1 (T1D) and type 2 (T2D) diabetes. Overall, the investigation is comprehensive and acceptable but results do not improve the predictive value presented. The author mentions that the high presence of diabetes duration in each model could have been masking the added predictive value of advanced variables. I suggest dividing diabetes duration and further statistical regression analysis, which may reduce its masking effect.
Author Response
I read with great interest the paper “Predictive Value of the Advanced Lipoprotein Profile and Glycated Proteins on Diabetic Retinopathy" by Julve et al.
The article is well written. Paper design is fine. The article is logically divided into sections and subsections.
We sincerely appreciate such overall favourable comments on our study's structure and experimental design.
Comments:
- Discussion: line 352-357: The authors have reported only HDL reduction linked to diabetic retinopathy. However, also high HDL cholesterol was found to be associated to DR (doi: 10.1016/j.diabres.2019.03.028). I also suggest the authors to perform such analysis in their manuscript.
We fully agree with the Reviewer’s comment. Indeed, elevated HDL was a significant feature in the DR group with T2D and, thus, support the notion raised by Carlo Sasso et al. (doi: 10.1016/j.diabres.2019.03.028). Noteworthy, HDL-C levels were also higher in advanced stages of DR in the T2D group.
Following the Reviewer’s suggestion, we further assessed the proportion of diabetic subjects (both T1D and T2D) with DR and in different DR stages in groups defined by three HDL-C concentration ranges (<30, 30-60, and >60 mg/dL). Our analysis showed that the proportion of T2D subjects with DR or advanced signs of DR was higher in the group of T2D having higher concentrations of HDL-C (>60 mg/dL). The proportion of T1D subjects with DR and or its severity grade did not differ among HDL-C categories. Of note, although the proportion of T2D subjects with HDL-C values below 30 mg/dL was higher, the number of subjects below this HDL-C cutoff was small (n=9), which might limit reaching any solid conclusion.
The reference suggested by the Reviewer has been added and commented in the Discussion section (lines 368-374, pages 11-12) in the final version of the revised manuscript. Additionally, the distribution of T2D subjects with DR in each HDL-C range is now in Supplementary Table S6.
- As we are assisting to a paradigm shift in diabetes treatment, in favour of a multifactorial intervention, I suggest the authors to report patients’ drug therapy (ezetimibe, antihypertensive, antidiabetics…). (doi: 10.1186/s12933-021-01343-1)
We thank the Reviewer for the suggestion. In this regard, our models already took into consideration the medications as they were run with the variables HT and DLP, which included by definition the subjects taking blood pressure lowering and lipid-lowering medications, respectively. The latter has been defined under the Material and Methods section (lines 107-119, page 3) in the revised version of the manuscript. Additionally, and in line with the Reviewer’s comment, now we have also prepared two new supplementary tables showing a list of medications that the study participants with T1D (Table S2) and T2D (Table S3) were receiving, respectively. The mention of drug therapies originally shown in Table 1 was moved from this table to either Table S2 or S3, as appropriate, in the final version of the revised manuscript.
Reviewer 2 Report
I read with great interest the paper “Predictive Value of the Advanced Lipoprotein Profile and Glycated Proteins on Diabetic Retinopathy" by Julve et al.
The article is well written. Paper design is fine. The article is logically divided into sections and subsections.
Comments:
1. Discussion: line 352-357: The authors have reported only HDL reduction linked to diabetic retinopathy. However, also high HDL cholesterol was found to be associated to DR (doi: 10.1016/j.diabres.2019.03.028). I also suggest the authors to perform such analysis in their manuscript.
2. As we are assisting to a paradigm shift in diabetes treatment, in favour of a multifactorial intervention, I suggest the authors to report patients’ drug therapy (ezetimibe, antihypertensive, antidiabetics…). (doi: 10.1186/s12933-021-01343-1)
Author Response
We appreciate the input given by the Reviewers, which enabled us to greatly improve the quality of our manuscript. In the following pages, we enclose our point-by-point responses to the Reviewers’ comments. Please, note that in the revised version of the Manuscript changes in response to the Reviewers’ comments are highlighted in yellow, so that they can be easily traced.
Reviewer 2
This study is to demonstrate the Prediction whether the advanced characteristics of serum lipoprotein subclasses could better predict the risk of developing diabetic retinopathy (DR) and its severity compared to other established risk factors in subjects with type 1 (T1D) and type 2 (T2D) diabetes. Overall, the investigation is comprehensive and acceptable but results do not improve the predictive value presented. The author mentions that the high presence of diabetes duration in each model could have been masking the added predictive value of advanced variables. I suggest dividing diabetes duration and further statistical regression analysis, which may reduce its masking effect.
We fully appreciate the Reviewer’s comments regarding the manuscript. Absolutely, diabetes duration is the main determinant in predicting both the presence of DR and its severity, and potentially mask the added value of other variables. As suggested, we categorized diabetes duration in tertiles; however, the predictive potential of advanced Liposcale variables was not significantly improved. As such this sub-analysis has been briefly mentioned and referred to as “data not shown” in the Results section (lines 263-266, page 8), and discussed in the Discussion section (lines 347-352, page 11). Remarkably, the upper and middle tertiles of diabetes duration remained as one of the main predictive variables of DR presence and severity, in line with what was already described in the initial models.
Reviewer 3 Report
Dear Editor,
I carefully read the manuscript by Julve and collaborators. Even though the study is interesting it does not match with the interests of the Journal.
My further minor comments are the following:
- Line 150: The authors should specify if they performed the Levene's test before the Student's T test.
- English language needs to be improved.
Author Response
We appreciate the input given by the Reviewers, which enabled us to greatly improve the quality of our manuscript. In the following pages, we enclose our point-by-point responses to the Reviewers’ comments. Please, note that in the revised version of the Manuscript changes in response to the Reviewers’ comments are highlighted in yellow, so that they can be easily traced.
Reviewer 3
Dear Editor,
I carefully read the manuscript by Julve and collaborators. Even though the study is interesting it does not match with the interests of the Journal.
My further minor comments are the following:
- Line 150: The authors should specify if they performed the Levene's test before the Student's T test.
We thank the Reviewer for pointing this out. We mistakenly named the test used; actually, we performed the Welch t-test, and not Student's. Thus, we there was no need to perform the Levene’s test. As the Reviewer already appreciated, the Student's t test is a special case of Welch's. Now, we have amended this error in the sub-section on statistical analysis of the Materials and Methods section.
- English language needs to be improved.
We thank the Reviewer's comment. The manuscript has been edited twice by an expert native English medical writer.
Round 2
Reviewer 2 Report
The authors fully answer to all the issues I raised. The reported new data on HDLc, supporting past evidence, has given more prominence to the article. The supplementary material is valuable.
Reviewer 3 Report
Dear Editor,
I carefully read the revised version of the manuscript. I still think that the topic of the present study does not match with the aim of the Journal. Then, it should not be published here.